# Enhanced CO$_2$ Methanation Reaction in C1 Chemistry over a Highly Dispersed Nickel Nanocatalyst Prepared Using the One-Step Melt-Infiltration Method

**Eui Hyun Cho [1], Woohyun Kim [2], Chang Hyun Ko [1],\* and Wang Lai Yoon [2],\***

[1] School of Chemical Engineering, Chonnam National University, 77, Yongbong-ro, Buk-gu, Gwangju 61186, Korea; bluejjaing@naver.com
[2] Hydrogen Energy Laboratory, Korea Institute of Energy Research, 152, Gajeong-ro, Yuseong-gu, Daejeon 34129, Korea; wkim@kier.re.kr
\* Correspondence: chko@jnu.ac.kr (C.H.K.); Wlyoon@kier.re.kr (W.L.Y.)

**Abstract:** The Paris Agreement requires the world to put the best efforts to reduce CO$_2$ emissions, due to the global warming problems. As a promising technology corresponding to this greenhouse gas treatment, the CO$_2$ methanation process a.k.a power to gas (PtoG), which catalytically converts CO$_2$ into methane, has been in the limelight. To develop an efficient catalytic process, it is necessary to design a low-cost and high-efficiency catalyst for high CO$_2$ conversion and CH$_4$ selectivity. In this study, we have developed Ni/γ-Al$_2$O$_3$ catalysts by the one-step melt-infiltration method, where both aging and calcination are done in one pot. For enhancement of the catalytic activity and selectivity, sufficient Ni content (>25 wt %) and a high dispersion (<10 nm) are simultaneously required. Thus, the aging conditions of the melt-infiltration methods, e.g., time and temperature, were optimized for the high dispersion with sufficient Ni content (15–50 wt %). The catalytic performance tests were carried out under atmospheric pressure, 275 to 400 °C and gas hourly velocity (GHSV) = 25,000 h$^{-1}$. And the various characteristic analyses (Brunauer–Emmett–Teller (BET), X-ray diffraction (XRD), H$_2$-chemisorption, temperature programmed reduction (TPR), etc.) were performed to confirm the effects on the catalytic performance. As a result, based on the experiments and the characterization data, the 30 wt %-Ni catalyst (Ni particles size = 11 nm) showed the best CO$_2$ conversion at 300 °C and the 20 wt % one having the highest Ni dispersion (Ni particles size = 8.8 nm), which showed the best intrinsic reaction rate and CH$_4$ selectivity in the entire temperature range.

**Keywords:** carbon dioxide; methanation; nickel catalyst; melt-infiltration method

## 1. Introduction

As part of the effort to ameliorate the problem of global warming, the share of renewable energy sources used to meet primary energy demands across the world reached around 10% in 2017 [1]. The increased amount of electricity generated from the increase in renewable energy in the power grid introduces issues of electricity infrastructure flexibility [2]. In countries, such as Germany and Denmark, where a large fraction of electricity is generated from renewable energy sources, various Research & Development, Business (R&BD) activities related to the power-to-gas technologies, in which excess electricity is converted into gas energy (hydrogen and/or methane) via electrolysis and methanation, are being carried out, to enhance the utilization of renewable energy sources, and the reliability of the power grid [3–7]. In particular, the CO$_2$ methanation reaction, which produces CH$_4$ via the reaction of H$_2$ and CO$_2$ over a catalyst, is being actively researched in Europe for the production of methane

for use in automobiles and heating [7]. The relevant $CO_2$ methanation process (the Sabatier reaction) is severely exothermic, and proceeds at 180–350 °C, at pressures of 1–100 bar. Therefore, managing the heat generated by the catalyst in fixed bed reactors and improving the reaction rate and methane selectivity are important in this reaction. As shown in Figure 1, $CO_2$ methanation reactions can be categorized based on their mechanism. In the reaction pathway (1), direct $CO_2$ methanation, $CH_4$ is formed directly without any intermediates; however, the direct conversion of $CO_2$ is very difficult, owing to its high activation energy. In indirect reaction pathway (2), $CO_2$ is first reduced to CO or HCOO; $CH_4$ is subsequently formed via the reduction of CO or HCOO by $H_2$ on the surface of the catalyst. Most of the previous studies on the $CO_2$ methanaton reaction have regarded reaction pathway (2) as the major $CO_2$ methanation pathway [8,9].

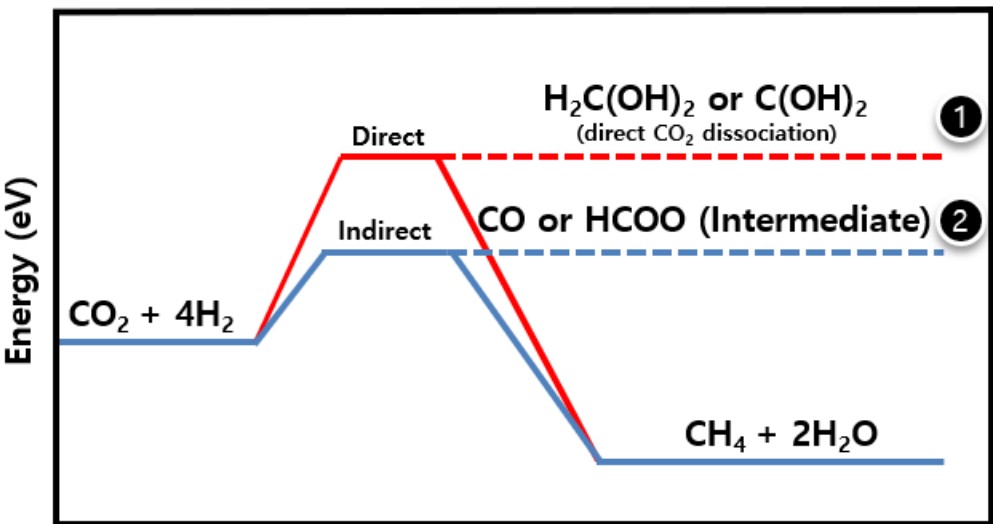

**Figure 1.** Potential reaction pathways in $CO_2$ methanation.

In the $CO_2$ methanation reaction mechanism, the rate-determining step is referred to as the CO decomposition reaction. To facilitate this step, $H_2$ must to be quickly decomposed by highly dispersed metal particles, and $CO_2$ should be easily adsorbed on the surface of the support [10]. Thus, a proper design of the catalyst to meet these requirements is important. First, the metal particles should be highly dispersed and have a size of 6–8 nm, and the metal loading should be higher than 30 wt %. Secondly, the surface of the support should contain oxygen vacancies to promote the adsorption of $CO_2$. Metallic catalysts that can be applied to $CO_2$ methanation reactions include Ni, Co, Fe, Ru, Rh, and +-Pd [11–17]. Among these metals, Ni is widely used, because of its relatively high activity and low cost, which are important for the development of economically feasible commercial processes Supports with mesoporous structures are often used to increase the diffusion rates of the reactants. Metal-oxide such as $Al_2O_3$, $TiO_2$, $Nb_2O_5$, $ZrO_2$, and $SiO_2$, can be used to improve $CO_2$ adsorption via oxygen vacancy [18–26].

Garbarino et al. investigated the effect of the Ni morphology of a $Ni/Al_2O_3$ catalyst on the $CO_2$ methanation reaction. They suggested that the Ni content should be sufficient to cover the surface area of the $Al_2O_3$ support, and that CO could be quickly converted to $CH_4$ via the strong interaction of cube-shaped Ni particles with the support [20]. Liu et al. reported that the high dispersion of Ni on catalysts synthesized via the deposition-precipitation (DP) method increased their $CO_2$ activity and $CH_4$ selectivity [21]. Muroyama et al. synthesized Ni supported catalysts on various oxide supports ($Y_2O_3$, $Sm_2O_3$, $ZrO_2$, $CeO_2$, $Al_2O_3$, and $La_2O_3$) to increase the $CO_2$ adsorption using oxygen vacancies [22].

The melt-infiltration method has been studied for the synthesis of catalysts with both high metal content and good metal particle dispersion. The simple synthesis process used in this method

does not increase the size of the metal particles, and the generation of unnecessary waste during catalyst synthesis is avoided, as this method does not require solvents that can dissolve metal salts. Melt-infiltration thus represents an alternative to the precipitation method, which has previously been used to produce catalysts with a high metal content and dispersion. It is applicable to catalysts for the Fisher-Tropsch reaction, nano-porous material, and template synthesis [27–31].

In this paper, we report on the synthesis of $Ni/Al_2O_3$ catalysts with a high nickel content and dispersion for the $CO_2$ methanation reaction, via the one-step melt-infiltration method. Mesoporous $Al_2O_3$ was used as the catalyst support. The one-step preparation of the catalyst was realized in an alumina crucible without the use of polypropylene bottles or an autoclave, which were used in previous studies. After optimizing the aging time, aging temperature, and calcination temperature, $Ni/Al_2O_3$ catalysts with Ni loadings of 15 wt %, 20 wt %, 30 wt %, 40 wt %, and 50 wt % were prepared. Using this method, the Ni particle dispersion was maximized, and $CO_2$ conversion and $CH_4$ selectivity were analyzed at a relatively high gas hourly velocity (GHSV) of ~25,000.

## 2. Results and Discussion

### 2.1. Effects of the Preparation Conditions on the Ni Dispersion

The one-step melt-infiltration synthesis method was used to prepare Ni catalysts with a high Ni content and dispersion. In this procedure, the nickel precursor was melted and subsequently inserted into the pores of the alumina support. An overview of the melt-infiltration procedure is presented in Figure 2. The thermal treatment conditions applied to the nickel precursors, namely the duration and temperature of the aging step, were expected to affect the size of the resulting nickel particles. © melted precursors are absorbed into the pores of the support by the capillary force, so the Ni loading was also expected to affect the final size of the Ni particle.

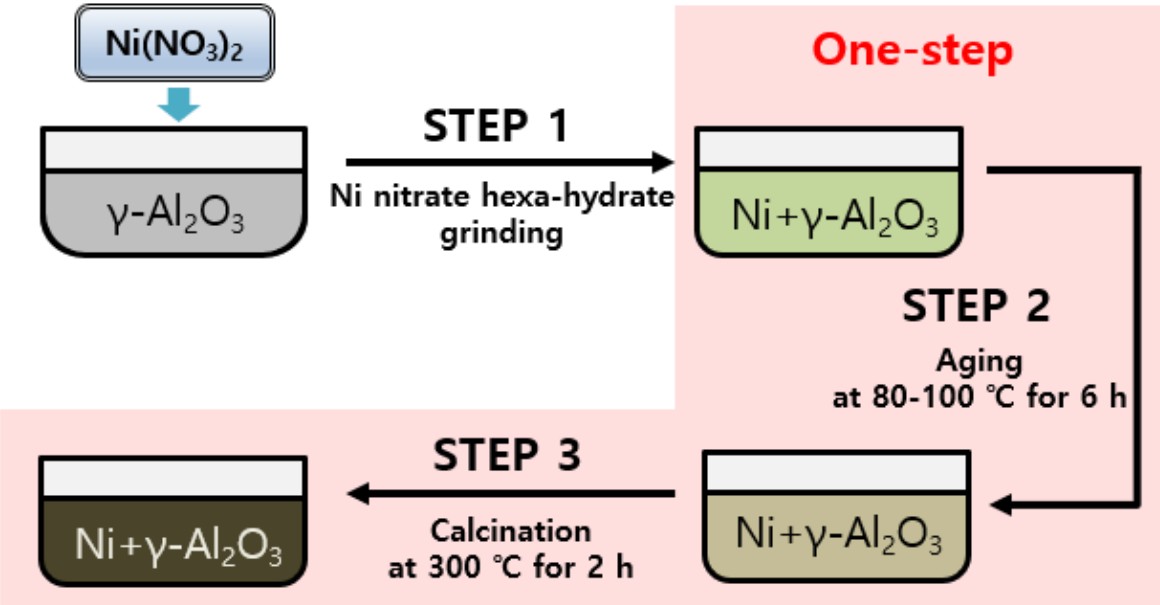

**Figure 2.** Overview of the experimental procedure.

The differential scanning calorimetry (DSC) analyses in Figure 3 show the changes in the nickel nitrate hexahydrate precursor with heating. All sample showed endothermic peaks at approximately 55 °C, 100 °C, 150 °C, and 275 °C. Based on the report of Brockner et al., [32] the peaks at 55 °C and 100 °C were attributed to water separation steps 1 and 2 [$Ni(NO_3)_2 \cdot 6H_2O \rightarrow Ni(NO_3)_2 \cdot 4H_2O \rightarrow Ni(NO_3)_2 \cdot 2H_2O$]. The melting point of nickel nitrate hexahydrate is 57 °C. The strong endothermic peaks at 55 °C can thus be assigned as the melting of the solid salts into liquids. The capillary forces,

which are the driving forces for the impregnation of the active components, seem to occur during water separation steps 1 and 2. Thus, achieving the high dispersion of Ni particles by optimizing the aging temperature is important. It was expected that the nickel salts would be impregnated mainly into the pores of the support at a sufficiently long aging time. As the nickel loading of the catalysts was increased, the peaks in steps 3 and 4 became larger. The peaks in step 3 were assigned as the decomposition of the precursor salts [$Ni(NO_3)_2 \cdot 2H_2O \rightarrow Ni(NO_3)(OH)_{1.50}O_{0.25} \cdot H_2O$]. Finally, the nitrate salt was completely decomposed and converted into nickel oxides between 250 °C and 300 °C in step 4 [$Ni(NO_3)(OH)1.50_{0.25} \cdot H_2O \rightarrow Ni_2O_3$ or $NiO$]. Capillary forces drive the impregnation of nickel nitrate in the melt-infiltration method, so the water separation steps 1 and 2 (temperature: 55–100 °C) would seem to be the most important. Thus, aging temperatures between 60 and 90 °C were investigated.

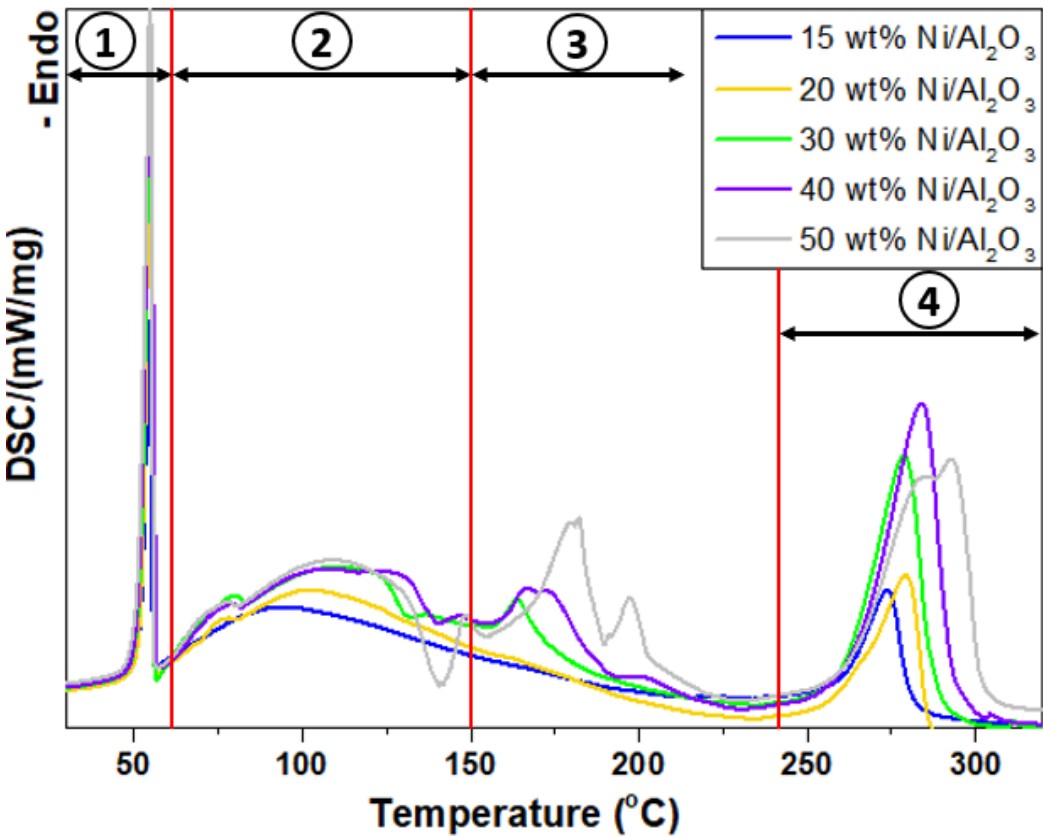

**Figure 3.** Decomposition of the nitrate salt at ~300 °C as measured by differential scanning calorimetry (DSC).

The size of the metallic Ni particles was minimized by adjusting the aging time, aging temperature, calcination temperature, and Ni loading. The size of the metallic Ni particles was determined using $H_2$-chemisorption, as detailed in Table 1 and Figure 4. The smallest Ni particle size was achieved using the following optimized conditions: an aging temperature of 80 °C and an aging time of 6 h with a fixed calcination temperature of 500 °C and a Ni loading of 20 wt %, as detailed in Figure 4a,b. To further reduce the particle size, the calcination temperature was reduced from 500 to 300 °C. The smallest particle size was obtained at 300 °C, as presented in Figure 4c. However, based on the DSC analysis, the calcination temperature should be higher than 300 °C, as the nitrate salts were not fully decomposed below 300 °C. Figure 4d shows the change in the size of the metallic Ni particles, depending on the Ni content (15–50 wt %); the catalysts were subjected to an additional "pre-decomposition" step, using the method of de Jong et al. [28] to further reduce the metallic Ni particle size. During the calcination step, the prepared samples were first heated at 100 °C for 3 h. The smallest Ni particle size of 8.8 nm

was achieved using nickel loading of 20 wt %. However, based on the metallic surface areas of Ni ($g_{cat}$) calculated using the total amount of adsorbed hydrogen (Table 1), the optimized Ni loading was estimated to be 30 wt %. The nickel surface area is closely related to the catalytic activity, with the higher $g_{cat}$ value corresponding to greater active sites area. Thus, the catalytic activity of the 30 wt % Ni catalysts was expected to be higher than those of the other catalysts, based on the number of active sites.

**Table 1.** Nickel particle sizes of the catalyst obtained using $H_2$- chemisorption.

| Property/Sample | 15Ni/Al$_2$O$_3$ | 20Ni/Al$_2$O$_3$ | 30Ni/Al$_2$O$_3$ | 40Ni/Al$_2$O$_3$ | 50Ni/Al$_2$O$_3$ |
|---|---|---|---|---|---|
| Metal dispersion (%) | 10.3 | 11.44 | 8.7 | 6.7 | 4.2 |
| Metal surface area ($m^2$/$g_{Cat}$) | 10.3 | 15.23 | 17.29 | 17.01 | 13.9 |
| Metal surface area ($m^2$/$g_{Ni}$) | 68.9 | 76.18 | 57.63 | 44.79 | 27.8 |
| Particle diameter (nm) | 9.8 | 8.8 | 11.7 | 15.0 | 24.2 |
| Nickel loading (wt %) [a] | 14.6 | 21.3 | 30.4 | 38.2 | 46.4 |

[a] Nickel loading was calculated by ICP-AES.

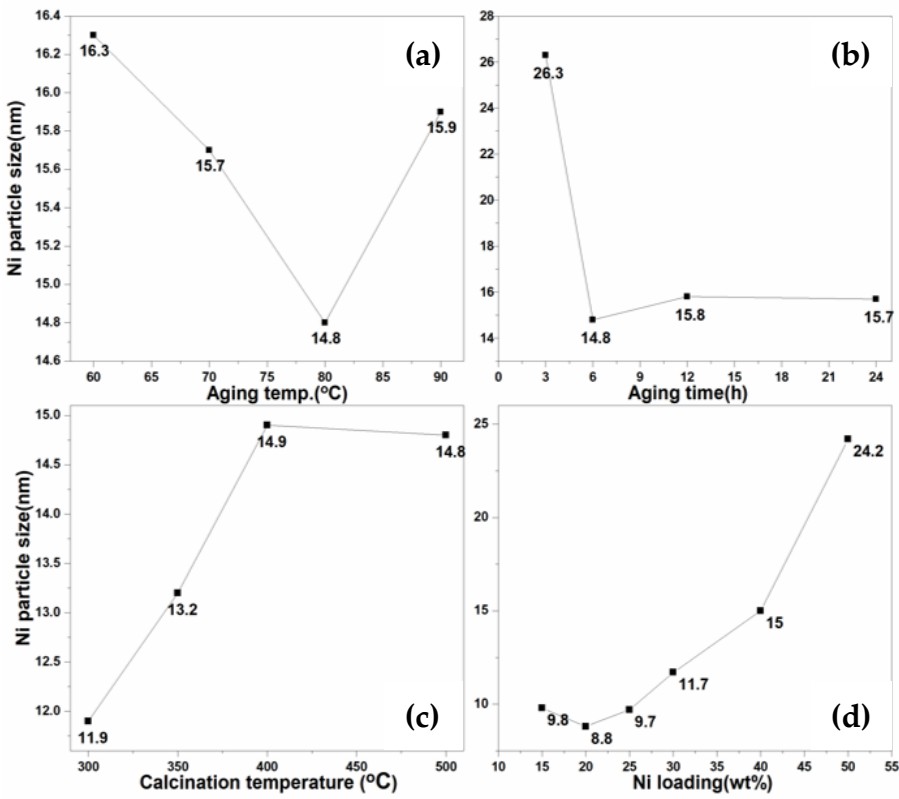

**Figure 4.** Effect of various synthesis conditions on the Ni particle size. (**a**) Effect of aging temperature investigated, using 30 wt % Ni/Al$_2$O$_3$ aging time of 6 h and a calcination temperature of 500 °C. (**b**): Effect of aging time on 30 wt % Ni/Al$_2$O$_3$, using an aging temperature of 80 °C and a calcination temperature of 500 °C. (**c**): Effect of calcination temperature on 30 wt % Ni/Al$_2$O$_3$, using an aging time of 6 h, an aging temperature of 80 °C and a calcination temperature of 300–500 °C. (**d**): Effect of Ni content on Ni/Al$_2$O$_3$ prepared under the optimal synthesis conditions (aging time: 6 h, aging temperature: 80 °C, calcination temperature: 300 °C).

The variation in the Ni particle size was also verified from their X-ray diffraction (XRD) patterns. As shown in Figure 5, the X-ray diffraction patterns of all the samples exhibited Ni peaks [44.7° (111), 51.7° (200), and 76.3° (220)] (JCPDS No. 04-0850) and their γ-Al$_2$O$_3$ diffraction peaks were observed at 39.5°, 46.1° and 66.9° (JCPDS No. 04-0850). Apparent metallic Ni peaks were observed for all

samples, except the Ni/Al$_2$O$_3$ catalyst with 15 wt % loading, owing to its well-dispersed small Ni particles. As the Ni loading was increased, the intensity of metallic Ni peaks increased (51.7° and 76.3°). The sizes of metallic Ni particles were calculated using the Scherrer equation. The main peak of metallic Ni particle was located at 45°, but this peak overlapped with the nearby Al$_2$O$_3$ peak. Thus, the well-resolved peak at 55° was used in the Scherrer equation. The Ni particle size of the 15 wt % Ni/Al$_2$O$_3$ catalyst was calculated to be 7.3 nm, 20 wt %: 8.0 nm, 30 wt %: 11.1 nm, 40 wt %: 16.1 nm, and 50 wt %: 20.4 nm. These results were in good agreement with the sizes of the Ni metal particles determined using the H$_2$-chemisorption analysis.

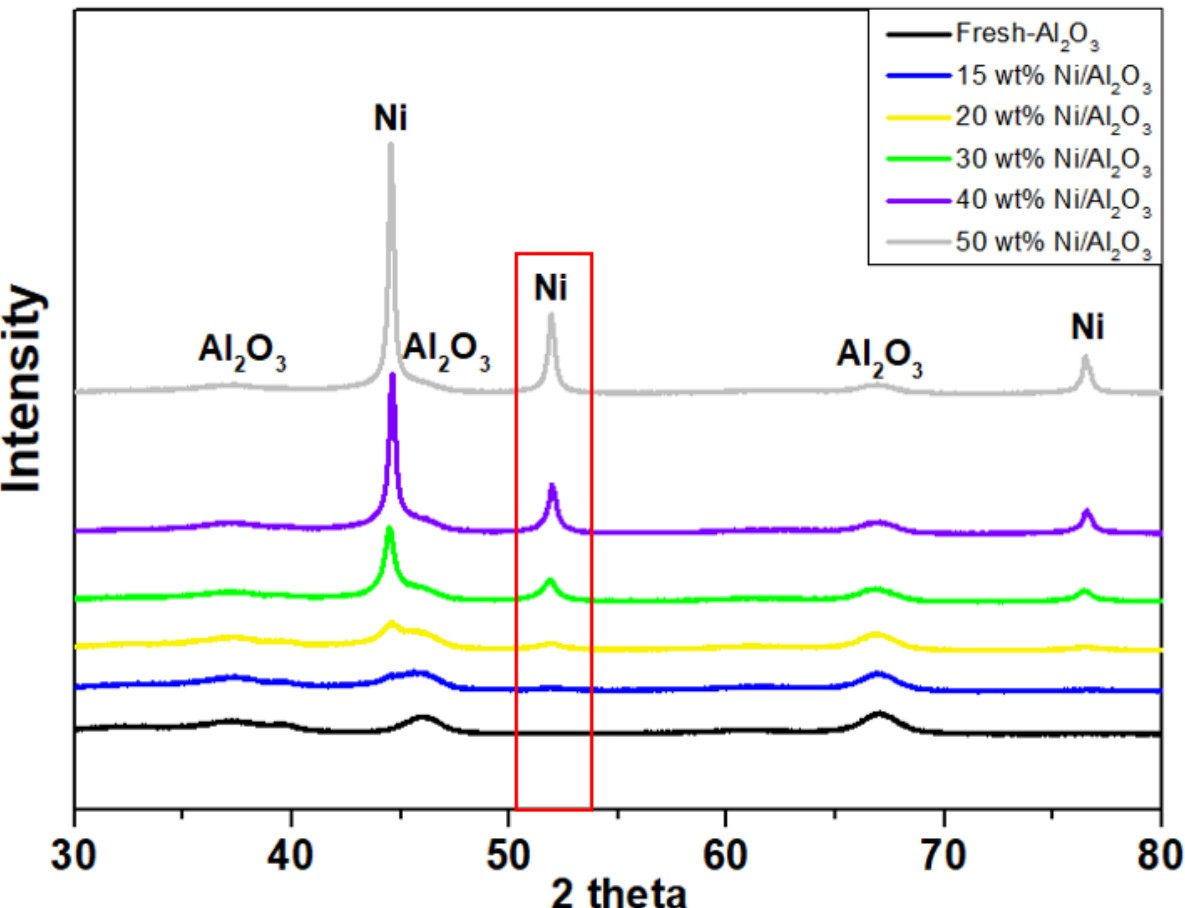

**Figure 5.** XRD patterns of the Ni/Al$_2$O$_3$ catalysts.

The metallic Ni particle sizes of the reduced 20 and 30 wt % Ni/Al$_2$O$_3$ catalyst were analyzed using a TEM. As shown in Figure 6a,b, the metallic Ni particles were suitably distributed over the supports. The sizes of the nickel particles were approximately 7–13 nm, which matched well with the metallic Ni particle sizes obtained using the H$_2$ chemisorption and XRD.

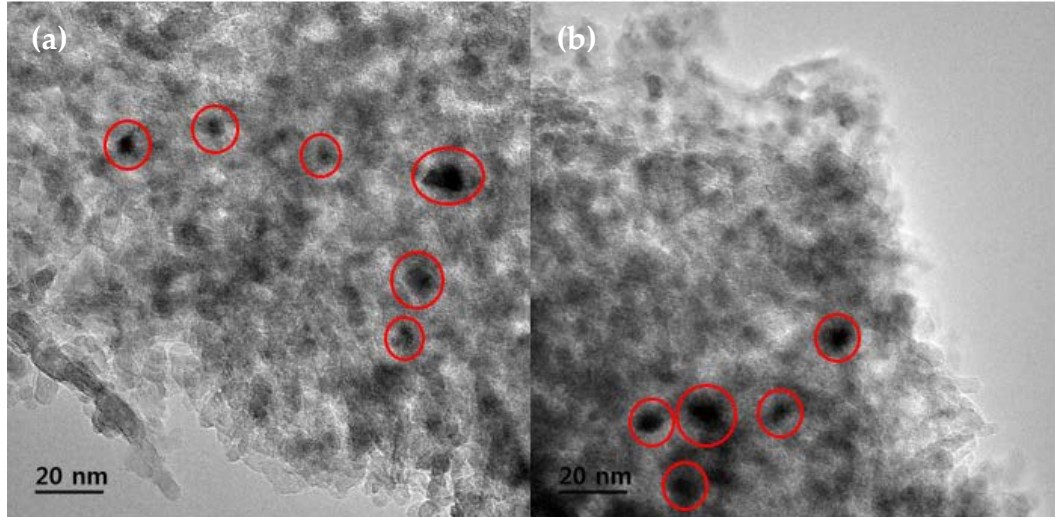

**Figure 6.** TEM images showing the Ni particle dispersion and size on the 20 wt % (**a**) and 30 wt % (**b**) catalysts.

## 2.2. Physical and Chemical Properties of Catalysts

The $N_2$ adsorption-desorption isotherms of the samples prepared by the melt-infiltration method are shown in Figure 7a. The surface areas and pore sizes of the catalysts were obtained using the Brunauer–Emmett–Teller (BET) and Barrett–Joyner–Halenda (BJH) analyses methods, and are detailed in Table 2 and Figure 7b. The isotherms were type IV with an H1 hysteresis loop according to the IUPAC classification, indicating that a mesoporous structure was maintained, even after a large amount of metallic nickel (e.g., 50 wt %) was incorporated. The mesopore volumes corresponding to pore diameters between 6 and 16 nm drastically decreased as the nickel loading increased. This indicated that most of the metallic Ni particles were likely located inside the mesopores. Among the five catalyst samples, the 15 wt %/$Al_2O_3$ catalyst exhibited the highest BET surface area The BET surface area decreased slightly with increasing Ni loading up to 30 wt %, and then decreased drastically at Ni loadings above 40 wt %. To increase the number of active sites by the formation of nickel particles with high dispersion, the physical surface area of the catalyst should be as large as possible.

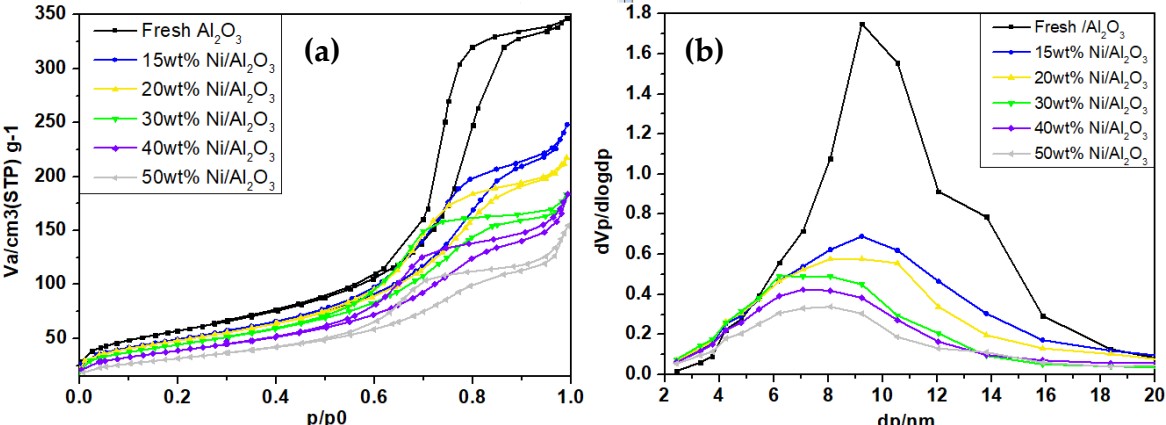

**Figure 7.** (**a**) $N_2$ adsorption isotherms, (**b**) pore size distributions calculated using the Barrett–Joyner–Halenda (BJH) method for the different catalysts.

**Table 2.** Physical properties of the nickel catalysts obtained using the nitrogen adsorption method.

| Property/Sample | Fresh-$Al_2O_3$ | 15Ni/$Al_2O_3$ | 20Ni/$Al_2O_3$ | 30Ni/$Al_2O_3$ | 40Ni/$Al_2O_3$ | 50Ni/$Al_2O_3$ |
|---|---|---|---|---|---|---|
| BET surface area ($m^2$/g) | 204.8 | 176.9 | 172.8 | 160.9 | 139.0 | 125.4 |
| Total pore volume ($cm^3$/g) | 0.5 | 0.4 | 0.3 | 0.3 | 0.3 | 0.3 |
| Average pre diameter (nm) | 10.5 | 9.2 | 7.7 | 7.0 | 8.0 | 8.7 |

Temperature programmed reduction (TPR) was used to investigate the reducibility of the catalyst; the results are presented in Figure 8. The free NiO particles, which were loosely bound to the support, gave rise to the first reduction peak at 200–220 °C [21]. As the temperature increased, NiO particles with medium-strength interactions, with the support formed second reduction peak at 260–400 °C. With increasing Ni loading, the area of the second reduction peak increased, and the peaks shifted to the lower temperature. The contact area of the metallic Ni particles with the support would be expected to decrease, as the size of the particles increased. The catalysts with low Ni loading (15 and 20 wt % Ni/$Al_2O_3$) showed a third reduction peak at high temperatures of around 450 °C. Thus, the high-temperature peak was attributed to the reduction of well-dispersed NiO particles, with strong binding to the alumina support. The catalysts with higher Ni loadings (30, 40, and 50 wt % Ni/$Al_2O_3$) did not exhibit distinct third reduction peaks, indicating the weaker interaction between the Ni particles and the alumina supports, owing to the larger size of the metallic particles. Additionally, the amount of adsorbed hydrogen was found to increase with the Ni loading.

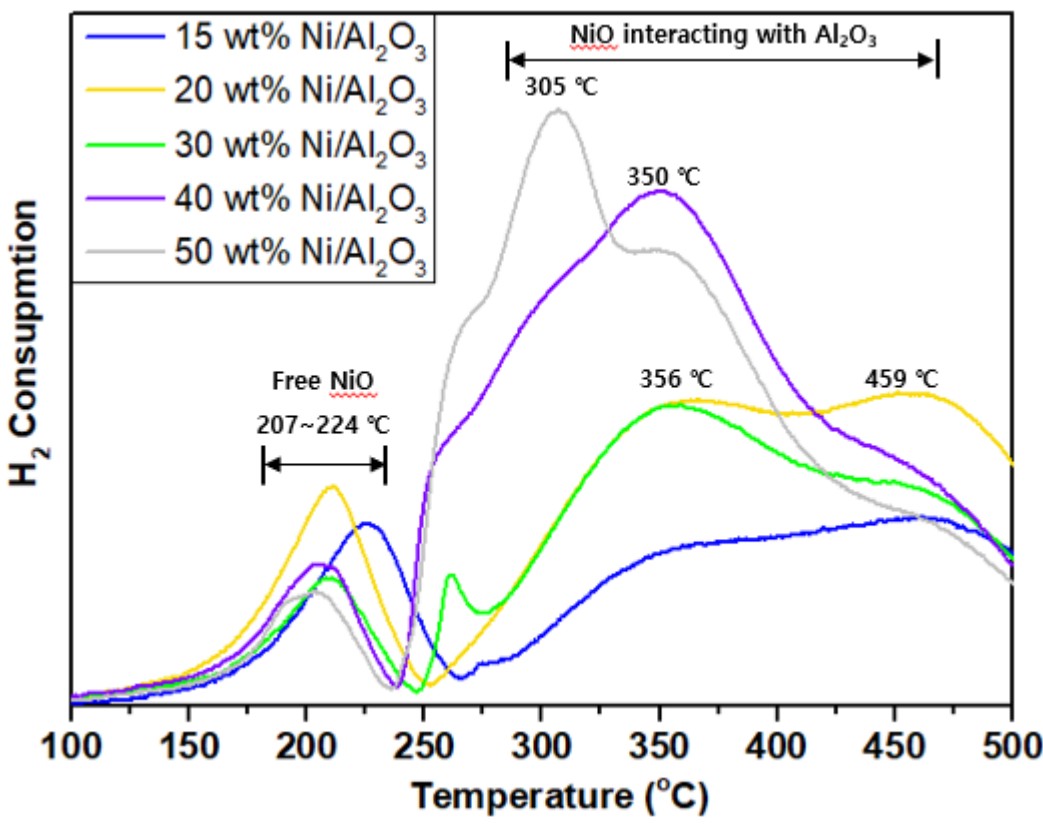

**Figure 8.** Temperature programmed reduction (TPR) profiles of the Ni catalysts.

*2.3. Catalytic Performance in the CO$_2$ Methanation Reaction*

Before the catalytic performance test, a blank test was conducted in the reaction temperature range of 275–400 °C and a GHSV of 25,000 h$^{-1}$; no CH$_4$ was detected. As shown in Figure 9, the catalyst

performance in terms of $CO_2$ conversion deteriorated as the GHSV was increased from 5000 to 45,000 $h^{-1}$, using the 20 wt % $Ni/Al_2O_3$ catalyst. To compare the catalytic performances of the given catalysts clearly, a GHSV of 25,000 $h^{-1}$ was selected, which represented the maximum space velocity when the pressure drop was limited to less than 1 atm.

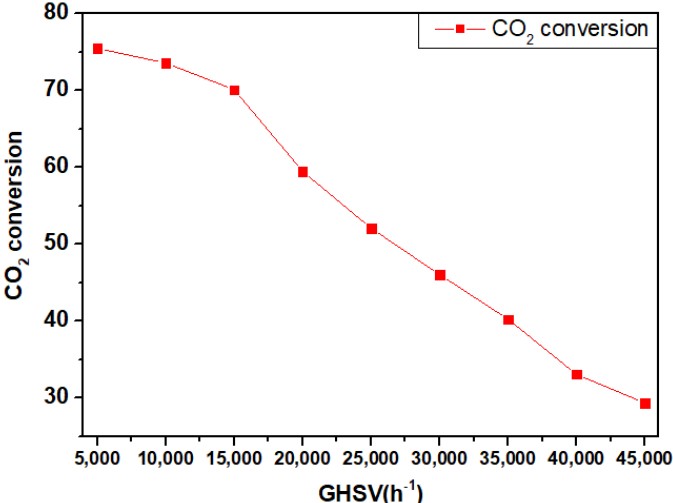

**Figure 9.** $CO_2$ conversion at different gas hourly velocity (GHSV) values ($h^{-1}$) at a reaction temperature of 300 °C, using 30 mg (original sample weight for activity test: 50 mg) of the 20 wt % Ni catalyst + 270 mg of SiC.

Figure 10a shows the changes in $CO_2$ conversion at different reaction temperatures in the $CO_2$ methanation reaction. The catalyst with a 20 wt % Ni loading (20 wt % $Ni/Al_2O_3$) was expected to exhibit the best catalytic performance, because of its smallest metallic Ni particle size and high BET surface area. However, the 30 wt % $Ni/Al_2O_3$ catalyst with a high metallic Ni surface area showed better $CO_2$ conversion than the 20 wt % $Ni/Al_2O_3$ catalyst at 300–400 °C. This indicated that the catalytic activity was closely related to the Ni metal surface area. Thus, to maximize the catalytic activity to $CO_2$ conversion, 30 wt % Ni loading was the best amongst the tested samples. The $CO_2$ conversions of the catalysts with high Ni loadings (30 to 50 wt %) were almost the same in the higher temperature region (350–400 °C), approaching thermodynamic equilibrium. In the meantime, the reference catalyst was less active than the other catalysts over the entire reaction temperature region, because it was not optimized for $CO_2$ methanation. The $CO_2$ conversions, of all the catalysts, decreased at high temperatures, owing to the reverse water gas shift reaction [7–9]. The low-Ni-loading catalysts (15, 20, and 30 wt % $Ni/Al_2O_3$), in which metallic Ni particles were highly dispersed, showed high $CH_4$ selectivity at low temperatures, as shown in Figure 10b. The 20 wt % $Ni/Al_2O_3$ catalyst especially, having the highest nickel dispersion, showed remarkably higher $CH_4$ selectivity, compared with the other catalysts. The $CH_4$ selectivity of all the catalysts also decreased with increasing reaction temperature owing to the reverse water gas shift reaction. In order to evaluate the intrinsic catalytic performance, the reaction rates for the conversion of $CO_2$ to $CH_4$ by the metallic Ni particles were calculated using Equation (3), and are presented in Table 3. The highest $CO_2$ methanation reaction rate was observed for the 20 wt % $Ni/Al_2O_3$ catalyst; the reaction rates followed this order: 20 wt % ($12.7 \times 10^{-2}$ $s^{-1}$) > 15 wt % ($9.1 \times 10^{-2}$ $s^{-1}$) ≥ 30 wt % ($8.8 \times 10^{-2}$ $s^{-1}$) > 40 wt % ($4.4 \times 10^{-2}$ $s^{-1}$) > 50 wt % ($3.1 \times 10^{-2}$ $s^{-1}$). These data provided further evidence that the intrinsic reaction rate for $CO_2$ methanation should be optimized by both Ni content and its dispersion.

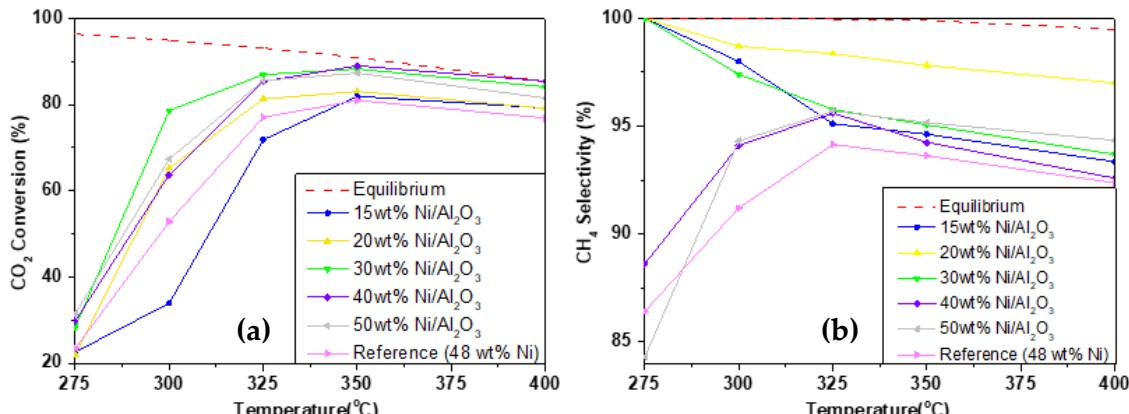

**Figure 10.** $CO_2$ conversion for the catalysts with different Ni loadings at different reaction temperatures (**a**) and $CH_4$ selectivities (**b**).

**Table 3.** $CO_2$ methanation reaction rate calculated using Equation (3).

| Sample | Rate $\times 10^{-2}$ ($mol_{CH4}$ $mol_{Ni}^{-1}$ $s^{-1}$) |
|---|---|
| 15Ni/Al$_2$O$_3$ | 9.1 |
| 20Ni/Al$_2$O$_3$ | 12.7 |
| 30Ni/Al$_2$O$_3$ | 8.8 |
| 40Ni/Al$_2$O$_3$ | 4.4 |
| 50Ni/Al$_2$O$_3$ | 3.1 |

Reaction conditions: T = 573 K, GHSV = 25,000 h$^{-1}$.

## 3. Materials and Methods

### 3.1. Materials

Unless otherwise noted, all chemicals were used as received without further purification. Gamma alumina powder (SASOL, Johannesberg, JHB, South Africa; mean pore diameter: 10 nm, BET surface area: 202 m$^2$/g) was used as a support, and nickel nitrate hexahydrate (Ni(NO$_3$)$_2$·6H$_2$O, 98%, JUNSEI, Tokyo, Japan) was used as the precursor. The prepared Ni catalyst samples were compared with the commercial Ni-based catalyst (48 wt % Ni, 3 wt % Ca, 6 wt % Mg/Al$_2$O$_3$). Based on its intended use in the CO$_2$ methanation reaction, a commercial catalyst designed for hydrogenation was selected as the reference catalyst. This reference catalyst was originally designed for use at a low space velocity.

### 3.2. Preparation of Nickel Catalysts

The melt infiltration method was employed to prepare highly dispersed nickel catalysts, following the procedure detailed in Figure 2. The alumina powder was ground in a mortar and sieved to a particle size of approximately 100 μm. The appropriate quantity of nickel nitrate hexahydrate salt to give the desired Ni content (15 wt %, 20 wt %, 30 wt %, 40 wt %, or 50 wt %) was ground together with the gamma-alumina powder (1 g) in the mortar. When the powder became homogeneously green, it was transferred to an alumina boat. Finally, in the aging procedure, the sample was slowly heated at a ramp rate of 0.7 °C/min, to a maximum temperature of 80 °C for 3 h, and then at 0.33 °C/min to a maximum of 100 °C for 3 h. Subsequently, the sample was calcined for 2 h, using a heating rate of 1.7 °C/min, and a maximum temperature of 300 °C.

### 3.3. Characterization of the Catalysts

Differential scanning calorimetry (DSC) analysis was used to confirm the melting period of nickel nitrate salt. This measurement was performed using a Q200 (TA Instruments, New castle, NW, United Kingdom), with a heating rate of 3 °C/min, and a maximum temperature of 400 °C under N$_2$ stream

(50 mL/min). The particle size of the nickel oxide was determined using X-ray diffraction (Rigaku, Tokyo, Japan). $H_2$-chemisorption analyses were also conducted to confirm the size and dispersion of the metallic particles, using an ASAP 2010 (Micromeritics, Georgia, GA, USA); for these analyses, approximately 50 mg of the catalyst was reduced under $H_2$ stream (200 mL/min) at 400 °C for 3 h, after which, $H_2$ adsorption measurement was performed under Ar stream (30 mL/min) at 50 °C. Temperature programmed reduction (Micromeritics, Georgia, GA, USA) was carried out, to identify the reduction temperature and the total $H_2$ consumption of the nickel catalysts. A 100 mg catalyst sample in a U-quartz cell was pre-treated with He gas at 400 °C for an hour, and then heated to 500 °C under $H_2$ stream (50 mL/min) at a heating rate of 5 °C/min. Brunauer–Emmett–Teller (BET) analysis was carried out using $N_2$ adsorption-desorption isotherms (Micromeritics, Georgia, GA, USA) to identify the surface area of each catalyst. Barrett–Joyner–Halenda (BJH) analysis was also conducted to determine the pore-size distribution and pore volume. A transmission electron microscope (JEOL, Tokyo, Japan) was used to confirm the size of the Ni particles.

### 3.4. Catalytic Activity Tests

The catalytic activity tests of the prepared samples and the reference catalyst for carbon dioxide methanation were performed at atmospheric pressure, using a feed gas composition of $CO_2$:$H_2$:$N_2$ = 4:1:5, and a GHSV of 25,000 $h^{-1}$ over a temperature range of 275–400 °C. The prepared catalyst was diluted with SiC (sample: 50 mg; SiC: 250 mg), and loaded in a fixed-bed quartz reactor (inner diameter = 4 mm). Before the reaction, the catalysts were reduced at 400 °C for 3 h under pure $H_2$ stream (200 mL/min), and then cooled to 250 °C. Subsequently, the activity test was performed at each temperature for 2 h to determine the stable activity and selectivity data. The product gas was analyzed using an online micro-gas chromatograph (INFICON, Bad Ragaz, Switzerland), equipped with a thermal conductivity detector (TCD). The carbon dioxide conversion and methane selectivity were calculated using the following equations:

$$X_{CO_2}(\%) = \frac{F_{CO_2,in} - F_{CO_2,out}}{F_{CO_2,in}} \times 100(\%) \qquad (1)$$

$$S_{CH_4}(\%) = \frac{F_{CH_4,out}}{F_{CO_2,in} - F_{CO_2,out}} \times 100(\%) \qquad (2)$$

$$Reaction\ rate = \frac{Number\ of\ molecules_{CH4}}{Number\ of\ molecules_{Ni}\ *\ Time} \qquad (3)$$

## 4. Conclusions

Nickel-based catalysts with high metal dispersion and high nickel loading were prepared by the one-step melt-infiltration method. In this method, the aging and calcination process were conducted consecutively to simplify the catalyst preparation processes. To achieve both high nickel loading and dispersion in the catalysts, the aging time and temperature were adjusted. DSC analysis indicated that a pre-decomposition step was required to further optimize the catalysts. The catalyst synthesis time was drastically reduced by using one-step heat treatment, rather than typical melt-infiltration methods. As a result of the high space velocity (GHSV of 25,000 $h^{-1}$), all the catalysts except 30 wt % Ni/$Al_2O_3$ exhibited inferior catalytic performances at low temperatures. However, at high temperatures of 325–400 °C, the catalysts with high nickel loading (30–50 wt %) showed high $CO_2$ conversion. The 15–30 wt % Ni/$Al_2O_3$ exhibited high $CH_4$ selectivity, owing to the dispersion effect of the small metallic Ni particles [14]. The 20 wt % Ni/$Al_2O_3$ had a higher intrinsic reaction rate than the 30 wt % Ni/$Al_2O_3$ catalyst, because of its small metallic Ni particles. However, 30 wt % Ni/$Al_2O_3$ showed the highest bulk reaction rate among the tested catalysts, because it provided the largest metal surface area (i.e., the highest number of active sites). Thus, 30 wt % Ni/$Al_2O_3$ was determined to be the optimum

catalyst for the $CO_2$ methanation reaction. The melt-infiltration method used in this study was useful for the simple preparation of metal-support catalysts with a high metal loading.

**Author Contributions:** Data curation, E.H.C. and W.K.; formal analysis, E.H.C.; funding acquisition, W.L.Y.; investigation, E.H.C.; methodology, E.H.C. and W.K.; supervision, C.H.K. and W.L.Y.; writing—original draft, E.H.C.; writing—review & editing, W.K. and C.H.K. All authors have read and agreed to the published version of the manuscript.

**Funding:** This work was conducted under the framework of the research and development program of the Korea Institute of Energy Research (C0-2406).

**Conflicts of Interest:** The authors declare no conflict of interest. All authors have read and agreed to the published version of the manuscript.

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
