# Peer review of "Enhanced CO2 Methanation Reaction in C1 Chemistry over a Highly Dispersed Nickel Nanocatalyst Prepared Using the One-Step Melt-Infiltration Method"

_catalysts, doi:10.3390/catal10060643_

Round 1

Reviewer 1 Report

The authors report an interesting work on developing better Ni/Al23 catalyst using one‐step melt‐infiltration method for CO2 methanation. Their catalysts outperform the reference catalyst, in terms of higher CO2 conversion and CH4 selectivity, with lower Ni loading. The authors also provide reasonable amount of characterization, such as XRD, TEM, ICP, N2 physisorption and TPR, to help the understanding of the catalysts. However, it is suggested that the authors could build better discussion on the catalytic performance with their characterization data, to draw better understanding on how different properties affect the catalysts. Overall, the work is interesting and I would recommend acceptance of the manuscript for publication with major revision.

  1. Data in Table 2 is exactly the same as Table 3. Also, the authors mentioned that metallic surface area in Table 2 is calculated from H2 chemisorption in line 130-131, but the caption of Table 2 says N2 Please correct.
  2. Is the rate be a function of phase of Ni? How does the phase of Ni change in the reaction with different temperature? The authors mentioned that the amount of adsorbed H2 increases with Ni loading. Do the authors have any number to support that?
  3. Usually, particle size increases with aging time but an opposite trend was shown in Figure 4b. Could the authors provide an explanation for that?
  4. The TPR result suggests that Ni stays in NiO in RT, so please be more careful in saying "metallic" Ni in the texts (e.g. line 163-166).
  5. The authors claimed that "To enhance the mass transfer of the reactants and products, the physical surface area of the catalyst should be as large as possible. " in line 180. However, it is not necessary that mass transfer is strongly correlated with support surface area. Please correct or elaborate this claim if the authors have data to support.
  6. Were the rates provided in Table 3 under strict kinetics control? From Figure 4, GHSV of 25000 corresponds to >50% of conversion. However, kinetics controls require <10% conversion and the authors should report the rate under such condition.
  7. The authors should provide reasons on why their catalysts were better than the reference catalyst. Was it simply because of smaller particle size?
  8. Please correct the legend in Figure 10b (it should be Al2O3 instead of Al2O).

Author Response

Pleases see the attachment

Reviewer 2 Report

The authors report on the development and performance of a Ni-alumina catalyst towards the methanation reaction. This is an important study that deserves publication in Catalysts. However, I would like the authors to clarify few queries.

Line 58, absorption should be adsorption.

Please include references to previous demonstration of the melt-infiltration method in preparation of Ni/Alumina catalysts.

What leads to the double peak structure in region 3 of DSC curve when the weight % is gradually increased?

Please add few sentences describing factors that could potentially hinder the migration of Ni ions inside Al2O3.

Was the catalyst stability tested for prolonged period? Please provide characterization results of the spent catalysts.

Please provide a tentative hypothesis of why the 30 wt % sample had the largest specific area.

Round 2

Reviewer 1 Report

The authors have made proper changes in the manuscript according to my suggestions, so acceptance of the paper is recommended.